behaviour, ecology

personality, breathing rate, cuckoo parasitism, egg rejection, Daurian redstart

**Author for correspondence:**
Wenhong Deng
e-mail: dengwh@bnu.edu.cn

# Host personality predicts cuckoo egg rejection in Daurian redstarts *Phoenicurus auroreus*

Jinggang Zhang[1,2], Peter Santema[2], Jianqiang Li[3], Lixing Yang[1], Wenhong Deng[1] and Bart Kempenaers[2]

[1]Ministry of Education Key Laboratory for Biodiversity Sciences and Ecological Engineering, College of Life Sciences, Beijing Normal University, Beijing, People's Republic of China
[2]Department of Behavioural Ecology and Evolutionary Genetics, Max Planck Institute for Ornithology, Seewiesen, Germany
[3]School of Ecology and Nature Conservation, Beijing Forestry University, Beijing, People's Republic of China

JZ, 0000-0002-0459-0429; JL, 0000-0002-0029-1747; WD, 0000-0002-2355-567X; BK, 0000-0002-7505-5458

In species that are subject to brood parasitism, individuals often vary in their responses to parasitic eggs, with some rejecting the eggs while others do not. While some factors, such as host age (breeding experience), the degree of egg matching and the level of perceived risk of brood parasitism have been shown to influence host decisions, much of the variation remains unexplained. The host personality hypothesis suggests that personality traits of the host influence its response to parasitic eggs, but few studies have tested this. We investigated the relationship between two personality traits (exploration and neophobia) and a physiological trait (breathing rate) of the host, and egg-rejection behaviour in a population of Daurian redstarts *Phoenicurus auroreus* in northeast China. We first show that exploratory behaviour and the response to a novel object are repeatable for individual females and strongly covary, indicating distinct personality types. We then show that fast-exploring and less neophobic hosts were more likely to reject parasitic eggs than slow-exploring and more neophobic hosts. Variation in breathing rate—a measure of the stress-response—did not affect rejection behaviour. Our results demonstrate that host personality, along the bold-shy continuum, predicts the responses to parasitic eggs in Daurian redstarts, with bold hosts being more likely to reject parasitic eggs.

## 1. Introduction

Obligate avian brood parasites lay their eggs into the nests of other species and thus transfer the costs of parental care to their hosts [1]. Consequently, hosts have evolved a variety of defences to reduce the incidence of parasitism and/or to minimize the negative fitness consequences of successful parasitism [2–9]. One widespread anti-parasite defence is the rejection of parasitic eggs from the nest [5,6,10]. Despite its effectiveness, hosts show variation in levels of egg rejection both within and among populations [8,11]. Understanding why some hosts reject brood parasitic eggs while others do not, remains challenging.

When deciding whether to accept or reject a potential parasitic egg, hosts have to balance the risk of mistakenly rejecting their own eggs against the cost of accepting a parasitic egg [12]. Empirical studies have shown several factors that are associated with between- and within- individual variation in egg-rejection behaviour in a host population. First, age and experience of the individual may play a role. For example, great reed warblers *Acrocephalus arundinaceus* that were older or had previous experience with being parasitized were more likely to reject a parasitic egg than young breeders or first-time hosts [13]. Second, rejection often depends on traits of the parasitic egg. For example,

experiments using artificial cuckoo eggs in ashy-throated parrotbills *Paradoxornis alphonsianus*, a frequent host of the common cuckoo *Cuculus canorus* (hereafter, cuckoo), showed that hosts were less likely to reject the parasitic egg when it was more similar to its own eggs [14]. Other studies showed that the rejection behaviour of the host depended on the perceived risk of brood parasitism [15]. For example, hosts were more likely to reject a parasitic egg when they observed a cuckoo near their nest [16,17]. Yet, a considerable amount of variation in egg-rejection behaviour among individuals typically remains unexplained.

Avilés & Parejo [18] proposed the host personality hypothesis, suggesting that the response to parasitic eggs depends on personality traits of the host. This may be adaptive if the risk of being parasitized differs between hosts with different personality. However, hitherto to our knowledge, no empirical evidence exists to support this hypothesis. Animal personalities are defined as consistent inter-individual variation in behavioural traits, such as activity, aggressiveness, boldness, neophobia and exploratory behaviour [19,20]. Different personality traits are often correlated with each other, such that suites of covarying traits form behavioural syndromes [21,22]. For instance, aggressive individuals also tend to be proactive, bold, risk-taking, less neophobic and fast-exploring [23–25]. A growing body of evidence suggests that personality traits can influence individuals in many aspects of their life history [20,26,27]. For example, exploratory behaviour has been related to individual survival [28], natal dispersal [29], extra-pair mating patterns [30], nest defence [31] and territory defence [32].

Empirical tests of a relationship between personality traits and egg-rejection behaviour in brood parasitized hosts remain scarce. The only direct test comes from a study on great reed warblers, in which the relationship between host aggressiveness and egg rejection was investigated, but no relationship was detected [33]. Several indirect lines of evidence suggest that personality traits could affect host egg-rejection behaviour. For example, egg rejection is based on a learning mechanism [34], whereby good learners would be better at discriminating parasitic eggs. Empirical evidence suggests that individual variation in discrimination learning is connected to individual variation in exploratory behaviour. For instance, in black-capped chickadees *Poecile atricapillus*, fast explorers learned acoustic cues more quickly [35], and similar results have been found in great tits *Parus major* and common starlings *Sturnus vulgaris* [36,37]. These results lead to the prediction that fast explorers would be better at rejecting parasitic eggs. On the other hand, slow-exploring black-capped chickadees learned to reverse previously learned natural category rules more quickly than fast explorers, suggesting that slow explorers may be more sensitive to environmental stimuli [38]. In this case, slow explorers are predicted to be better at egg-rejection behaviour. Moreover, since egg-rejection behaviour comes with potential costs (e.g. recognition error and revenge by the parasite [12,39]), shy hosts may be more likely to tolerate or accept the brood parasitism to avoid these costs, while bold individuals may be more likely to take the risk and reject the parasitic egg. Because they are generally more aggressive, bold hosts may also be more effective than shy individuals at driving brood parasites away from their nest [31]. Being more successful at earlier lines of host defence may decrease selection on later lines of defence [40,41]. Thus, bold individuals may be better at keeping brood parasites away from their nest, but less efficient at discriminating and ejecting the parasitic egg, as they may have less opportunity to learn to reject a parasitic egg [18], but see [33]. On the other hand, shy individuals may be less frequently parasitized, for example, because they are less active and thus less conspicuous than bold individuals [42,43], and they may therefore also experience less selection to be good egg rejecters [18].

We report on an experimental study to test the host personality hypothesis using the Daurian redstart *Phoenicurus auroreus* as a model species. Daurian redstarts are a common cuckoo host and individuals of this species vary in their responses to parasitic eggs [44]. Males never reject a parasitic egg, but about half of the females do, while the other half accept such an egg [44]. Thus, the Daurian redstart provides an ideal system in which to investigate variation in egg-rejection behaviour [44]. The species also shows an egg colour polymorphism, with some females laying blue and others pink eggs, whereby the latter are more distinct from the blue cuckoo eggs [44,45]. The egg colour polymorphism is often interpreted as an adaptation against brood parasitism [14,46]. In our study population, redstarts laying blue eggs appear to suffer higher risk of parasitism than hosts laying pink eggs, although unbiased information on parasitization rate of the latter hosts are lacking as they may have ejected the parasitic egg before we detect it [44].

We previously found that egg-rejection behaviour in Daurian redstarts varied with host clutch colour and with the risk of parasitism: females laying pink eggs were more likely to reject foreign eggs than individuals laying blue eggs, and hosts experiencing a higher risk of being parasitized (cuckoo presence, see below) had higher egg-rejection rates [44]. However, a lot of variation in egg rejection among females remains unexplained. Therefore, we explored whether host personality traits affect egg-rejection behaviour. We first tested whether Daurian redstarts showed consistent inter-individual variation in two personality traits (exploration and neophobia) and in a physiological trait (breathing rate) across time. Second, we show how the two personality traits and breathing rate are correlated. Lastly, we investigated how these traits influence the response of the host to a parasitic egg.

## 2. Methods

### (a) Study system and general procedures

We studied a population of Daurian redstarts in the village of Shuangyu in Jilin, northeast China (43°37′19″ N and 126°09′54″ E) in 2019 and 2020. The study site is about 50 ha, and contains 170 nest-boxes. In our study site, females start laying from mid-April onwards and typically produce at least two clutches within one breeding season (clutch size (mean ± s.d.): $6.4 \pm 0.6$ ($n = 99$) and $5.5 \pm 1.0$ ($n = 163$) in the first and second clutch, respectively). Cuckoos arrive at the breeding grounds around mid-May (13 May in 2019), when most hosts have nestlings or are in the late incubation stage of their first clutch. Thus, in this population of Daurian redstarts, the risk of cuckoo parasitism varies within each breeding season from zero in the first clutch to a high risk in subsequent clutches (for detailed information, see [44]). Cuckoo eggs in Daurian redstart nests are pale blue with or without thin brown lines; they mimic the blue morph of host eggs, but are paler and bigger [44].

During each breeding season, we searched for natural nests every day and checked nest-boxes every week. When cuckoos

were present at the study site, we checked active nests (natural or in a nest box) every 1–2 days to assess whether it contained a cuckoo egg. We followed a total of 577 redstart nests, 370 in 2019 and 207 in 2020. Of these, 67 were naturally parasitized by a cuckoo egg, 43 in 2019 and 24 in 2020.

## (b) Experimental procedure

To assess egg-rejection behaviour of Daurian redstart females, we performed a brood parasitism experiment using a real cuckoo egg or a model egg that mimics a real cuckoo egg. We manufactured model cuckoo eggs using clay and painted them with acrylic colours. Mass and size of the model cuckoo eggs were similar to real cuckoo eggs [44]. We performed a total of 97 trials, 54 in 2019 and 43 in 2020. For each trial, we introduced a model cuckoo egg into the focal nest, during either the late-laying phase of the host, i.e. when the nest contained three eggs, or during early incubation, i.e. within 3 days after clutch completion. After artificially parasitizing a nest or after finding a naturally parasitized nest, we checked it daily for 6 days to decide the fate of the parasitic egg. We considered the experimental egg 'accepted' when it was still present in an active nest 6 days after it was introduced and 'rejected' when it disappeared while the nest was active and the host clutch was not reduced. Out of 67 nests that were naturally parasitized, we had personality data for 18 host females (eight in 2019 and 10 in 2020). Only these 18 nests were therefore included in the analysis.

We considered the nest to be deserted when the parasitic egg was still present, but the host had abandoned the nest within 6 days [47,48]. Our previous work showed that nest desertion rates did not differ between experimentally parasitized and non-manipulated control nests [47]. We therefore assumed that nest desertion in this study was not a consequence of the (artificial) parasitism and excluded deserted nests from further analysis.

## (c) Personality assays

### (i) Exploratory behaviour

We tested exploratory behaviour using the novel-cage approach described in Kluen et al. [49] and validated in other passerines [50,51]. The exploration cage was adapted from a wooden box (L 60 × W 40 × H 80 cm), fitted with six perches (25 cm) and one mesh side, and connected to a small metal compartment (L 20 × W 20 × H 20 cm) (electronic supplementary material, figure S1). During the 2019 and 2020 breeding season, when nestlings were 4 days old (hatching date = day 0), we caught adults by one of four methods (i.e. mist-net, tuck net, spring net traps or bird glue; see the electronic supplementary material). After banding, each individual was kept in the small compartment for 10 min of acclimatization. Then, the bird was released into the exploration cage through the connecting door without handling, and its behaviour recorded with a video camera, placed 3 m in front of the exploration cage, for 2 min. We scored exploratory behaviour as the number of hops within a location plus the number of movements (flights or hops) between different locations, including two floor sections and six sections within the cage area (scores ranged from 2 to 151 [51,52]).

### (ii) Neophobia

Neophobia is commonly measured as the reluctance of individuals to return to a known resource in the presence of a novel object [53–56]. In this study, we measured the female's latency to return to her nest (nest-box or natural nest) in the presence of the novel object during the incubation stage. During the 2020 breeding season, we performed novel object and control tests between 9 and 11 days after the start of incubation. We conducted experiments only in the afternoon (15.00–18.00), and only when the female was on the nest. First, we induced the female to leave by tapping the nest-box or the foundation of the natural nest. Then, we placed a yellow or red ping-pong ball on top of the nest-box or 10–15 cm above the natural nest and a video camera 5 m from the focal nest [55,56]. We then recorded the nest (box) for 60 min and measured return latency (in minutes). To confirm that the observed responses were caused by the novel object rather than by human disturbance, we conducted control trials, following the same procedure but placing no novel object near the nest. For 43 females that had undergone the artificial brood parasitism experiment, we conducted a first novel object test (yellow ball) and a control test. For 27 of these females, we performed a second novel object test using a red ball to determine the repeatability of the neophobia response.

We performed the tests (control, yellow ball and red ball) on three consecutive days. To avoid order effects, we performed the first novel object (yellow ball) and control tests in a randomly determined order (days 9 and 10 of incubation). The second novel object test (red ball) was performed on the third day (day 11).

### (iii) Breathing rate

Breathing rate is a physiological trait that has been proposed as an indicator of the stress response in songbirds [57]. Immediately after capture, we measured breathing rate by counting the number of breast movements within 30 s, following [58].

## (d) Statistical analyses

As defined, a meaningful personality trait should be individually repeatable [22]. We therefore tested the adjusted repeatability of all three measured traits using linear mixed-effects models (LMMs) with the trait measure as the dependent variable and bird identification as the random effect, using the R package rptR [59,60]. Following recommendations of Nakagawa & Schielzeth [59], we retained individuals with only one measure in the models. For breathing rate and exploratory behaviour (number of movements), we included sex, capture method, the date on which the test was performed (day of the year), test sequence, and the interval between two tests (in days) as fixed effects. For the neophobia response, we included return latency (during novel objects trials) as the dependent variable, with object type (yellow ball/red ball), nest type and baseline return latency as fixed effects. To avoid possible model overfitting, we further ran LMMs to detect what variables explained the significant variation in the personality trait. We then recalculated the repeatability of the three traits again, only including variables that explained significant variation in the personality trait. This approach gave qualitatively the same results (see the electronic supplementary material). We log-transformed data of return latency to meet the normality assumption.

To determine whether the return latency during the novel object treatment was a neophobia response rather than a response to human disturbance, we used a Wilcoxon matched-pairs test to compare the return latency between the control and the first novel object trial (yellow ping-pong ball).

We calculated a Pearson correlation coefficient between breathing rate and exploratory behaviour (number of movements), and a Spearman's rank correlation coefficient between return latency and either exploratory behaviour or breathing rate.

We used generalized linear models (GLMs) with a binomial error structure to examine whether the hosts' personality traits explained their response towards a parasitic egg. In all models, we included the response to the parasitic egg (rejected/accepted) as the dependent variable, and the three measured traits, clutch colour (blue/pink) and cuckoo egg (real/model) as explanatory variables. When testing the effects of breathing rate and exploratory behaviour on egg rejection, we also included cuckoo status (present/absent) as an explanatory variable. In 2020, we started the fieldwork only in late May (because of the COVID-19 pandemic), when cuckoos had already arrived at the study site.

Thus, the novel object experiments were all conducted during cuckoo presence. Therefore, we did not include cuckoo status as a fixed effect when examining the effect of return latency on hosts' response to the parasitic egg.

We used an information-theoretic approach to establish a candidate set of all possible models, and selected the best-fit model by comparing the corrected Akaike's information criterion (AICc). First, we selected a subset of models with the δ AIC value (ΔAICc) lower than 2 [61,62]. Then, we chose the most parsimonious model (i.e. the one with the smallest number of parameters) from this subset [63]. We calculated the total explanatory power of the model using Nagelkerke's $R^2$ (R package *fmsb*) [64], and the explanatory power of the parameters retained in the final model was assessed using hierarchical partitioning (R package *hier.part*) [65]. We also evaluated multicollinearity using the all variance inflation factor (VIF) in the final models. All VIF values were lower than 2, indicating weak correlation between the explanatory variables [66].

Some individuals were subjected twice to the breathing rate measurement, exploration test and parasitic egg experiments. When calculating correlation coefficients and running GLMs, we only used the data of the first measurement or test for these individuals. All statistical analyses were conducted in R 3.4.2 [67].

## 3. Results

### (a) Personality traits

Females took significantly longer to return to the nests during the first novel object trials than during the control trials ($V = 129$, $p < 0.001$, $n = 43$), indicating that females showed a neophobia response to the novel object. The return latencies were repeatable across the novel object trials ($r = 0.78$ (0.63, 0.90), $p < 0.0001$, $n = 70$ observations on 43 females), indicating that the neophobia response is a consistent personality trait. Both breathing rate and exploratory behaviour (number of movements) were repeatable across years in Daurian redstarts (breathing rate: $r = 0.53$ (0.39, 0.78), $p < 0.001$, $n = 343$ observations on 307 birds; exploratory behaviour: $r = 0.31$ (0.10, 0.68), $p = 0.03$, $n = 343$ observation on 305 birds).

There was a negative relationship between exploratory behaviour and return latency ($r_s = -0.54$, $p < 0.001$, $n = 43$; figure 1), i.e. fast-exploring females returned faster to their nests when a novel object was present than slow explorers. However, breathing rate was not significantly related to either return latency ($r_s = 0.02$, $p = 0.90$, $n = 43$), or to exploratory behaviour ($r_p = -0.04$, $p = 0.47$, $n = 301$).

### (b) Personality and egg rejection

Of the 18 redstart nests that were naturally parasitized by cuckoos in our population, five females rejected the egg, 11 females accepted the egg and two nests were predated during the experiment. Out of 97 nests that were artificially parasitized, 50 females rejected the egg, 45 females accepted the egg and two nests were predated during the experiment.

The models including the neophobia response, exploratory behaviour and stress response had an $R^2$ value (total variation in egg rejection explained) of 0.30, 0.42 and 0.35, respectively. The neophobia response (return latency) was the only significant predictor in the model ($p = 0.005$; table 1). Exploratory behaviour was also an important predictor, making up 25.77% of the explained variance ($p = 0.012$; table 1). The stress response (breathing rate) did not predict egg-rejection behaviour, and was excluded from the model (table 1). The

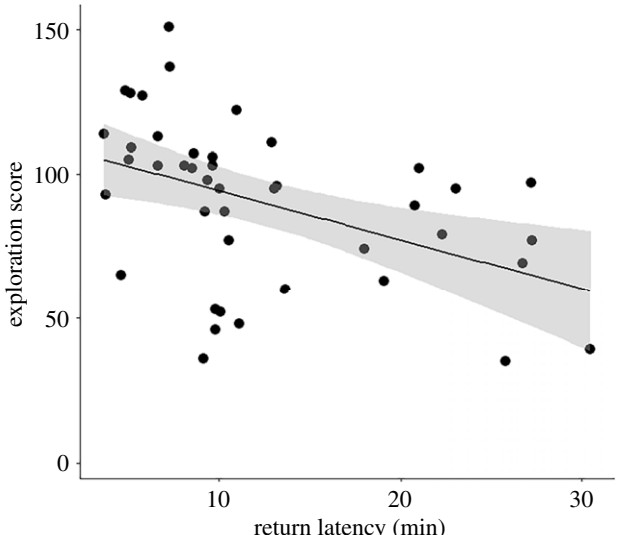

**Figure 1.** The relationship between exploratory behaviour (number of movements in the novel cage) and neophobia response (return latency). The grey shading indicates the 95% confidence intervals. Only data from the first novel object trial are shown ($n = 43$ females).

neophobia response and exploratory behaviour are thus important and significant predictors of egg-rejection behaviour in Daurian redstarts, with females that are less neophobic and more explorative being more likely to reject the parasitic egg (figure 2).

To test the robustness of our finding that egg-rejection behaviour was related to personality type, we additionally performed a Wilcoxon signed-rank test to compare the return latency and exploration score, and a Student's *t*-test to compare the breathing rate, between females that rejected the egg and females that accepted the egg. Females that rejected the egg indeed had a shorter return latency (mean ± s.d.: $9.6 \pm 5.4$ min, $n = 26$) than females that accepted the egg ($16.8 \pm 8.1$ min, $n = 17$; $V = 357$, $p < 0.001$). Females that rejected the egg also had a higher exploration score ($80.6 \pm 34.7$, $n = 55$) than females that accepted the egg ($59.3 \pm 22.7$, $n = 56$; $V = 936.5$, $p < 0.001$). However, females that rejected the egg had a similar breathing rate ($171.7 \pm 32.8$ min$^{-1}$, $n = 54$) as females that accepted the egg ($176.0 \pm 28.1$ min$^{-1}$, $n = 56$; $t = 0.74$, $p = 0.46$).

## 4. Discussion

This study shows that personality traits of a common host predict the host's response to a brood parasitic egg. Specifically, we show that fast-exploring and less neophobic hosts were more likely to reject parasitic eggs than slow-exploring and more neophobic hosts. We did not find a significant effect of breathing rate, a measure of the stress response of an individual, on host egg-rejection behaviour. We further demonstrate that object neophobia and exploration are repeatable across tests or across years in Daurian redstarts, and thus represent personality traits. Breathing rate was also highly repeatable across years, suggesting that it is a reliable indicator of the stress response in this species. Besides host personality, we found that host clutch colour (reflecting the degree of similarity between host and foreign eggs), the type of parasitic egg and cuckoo presence (perceived risk of brood parasitism) had significant effects on the female's

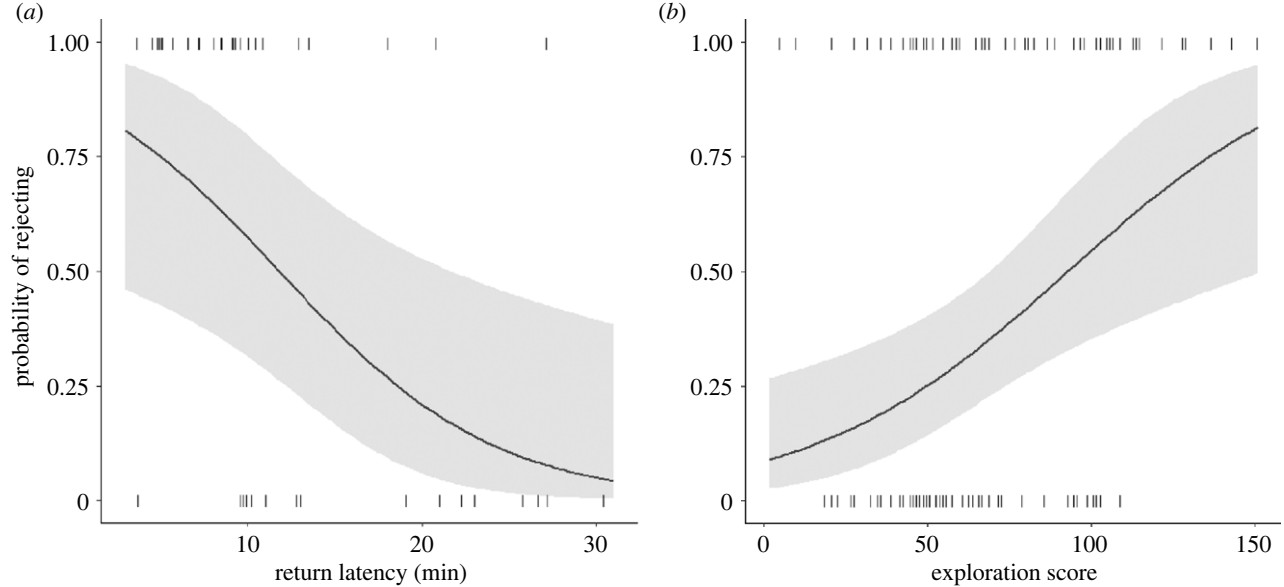

**Figure 2.** Relationships between the probability of rejecting the parasitic egg and (a) return latency and (b) exploration score (number of movements in novel cage) of the host. The grey shading indicates the 95% confidence intervals from the GLM. Tick marks indicate raw data points.

**Table 1.** Generalized linear models predicting the probability that a female Daurian redstarts rejected a parasitic egg. (The presented models are the most parsimonious models with a δ AICc lower than 2 (electronic supplementary material, table S2). The full models included the personality trait of interest, egg colour (blue or pink), cuckoo egg type (real or model) and cuckoo status (present or absent). For each fixed effect, the reference category is indicated in parentheses. *I* (%) is the proportion of the total variance explained by the models. VIF indicates the variance inflation factor for each predictor.)

| personality trait | fixed effect | estimate | 95% CI | Z | p-value | I (%) | VIF |
|---|---|---|---|---|---|---|---|
| neophobia response | intercept | 5.44 | 2.22–9.49 | 2.99 | 0.003 | | |
| | return latency | −1.90 | −3.71 to −0.78 | −2.47 | 0.005 | 100 | – |
| exploratory behaviour | intercept | −2.99 | −4.60 to −1.62 | −3.97 | <0.001 | | |
| | exploration score | 0.02 | 0.01–0.04 | 2.50 | 0.012 | 25.77 | 1.06 |
| | clutch colour (blue) | 1.96 | 0.97–3.06 | 3.71 | <0.001 | 43.79 | 1.14 |
| | cuckoo egg type (model) | −1.64 | −3.36 to −0.18 | −2.07 | 0.039 | 13.55 | 1.07 |
| | cuckoo status (absent) | 1.40 | 0.35–2.54 | 2.54 | 0.011 | 16.89 | 1.22 |
| stress response | intercept | −1.71 | −2.77 to −0.81 | −3.46 | <0.001 | | |
| | clutch colour (blue) | 1.95 | 1.01–3.01 | 3.87 | <0.001 | 55.11 | 1.13 |
| | cuckoo egg type (model) | −1.70 | −3.37 to −0.30 | −2.23 | 0.026 | 17.38 | 1.07 |
| | cuckoo status (absent) | 1.72 | 0.73–2.83 | 3.26 | 0.001 | 27.52 | 1.20 |

egg-rejection decision (table 1), consistent with our previous findings [44]. Together, these variables explained 30–42% of variation in egg-rejection behaviour. This implies that there is still unexplained variation in egg-rejection behaviour among Daurian redstarts that warrants further investigation.

## (a) Repeatability of personality traits

Exploratory behaviour is a commonly studied personality trait in both captive and wild bird populations [24,36]. In this study, we show that exploration of a novel cage was individually repeatable over years, also in Daurian redstarts. The repeatability value was similar to that reported in other species [28,51].

Most studies on neophobia as a personality trait were conducted with captive birds, but a few studies used individuals of free-living populations. [55,56,68]. Here, we show that the estimate of neophobia (return latencies between the first and second novel object trials) were repeatable in Daurian redstarts.

Therefore, our study provides clear evidence supporting that neophobia is a personality trait in this natural population.

## (b) Correlations between personality traits

We detected a negative correlation between exploratory behaviour and neophobia in Daurian redstarts, similar to patterns found in other species [55,56]. Exploratory behaviour, neophobia and boldness are inter-related, and form a bold-shy continuum [28,69], which reflects the best-studied personality axis in non-human animals [70,71]. Empirical evidence suggests that bold individuals tend to be proactive, fast-exploring and risk-taking (less neophobic), while shy individuals are reactive, slow-exploring and risk-averse (more neophobic) [70,72].

We did not find significant relationships between the stress response (breathing rate) and either exploratory behaviour or neophobia. This result is in line with a study

on great tits, where no significant correlations were detected between breathing rate and exploration in both city and forest populations [73].

## (c) Host personality predicts the response to a parasitic egg

According to the host personality hypothesis, personality traits of the host may influence their anti-parasite defence behaviour [18]. The first supportive evidence came from a study on the great reed warbler, showing that more aggressive females (during handling after being caught) had higher levels of nest defence (aggression towards cuckoos), but no relationship between host aggressiveness and egg rejection behaviour was detected [33]. In this study, we show that fast-exploring and less neophobic female hosts were more likely to reject a parasitic egg than slow-exploring and more neophobic females. A study on great reed warblers showed that females which devoted more time to clutch inspection ejected experimental eggs more quickly than hosts inspecting their parasitized clutches only briefly [74], which may at least indirectly suggest a positive relationship between host exploratory behaviour and egg rejection. According to the bold-shy continuum, our results suggest that bold hosts are more likely to reject parasitic eggs than shy individuals, which contrasts with the prediction from Aviles & Parejo [18].

One adaptive explanation regarding the existence of the bold-shy continuum is that bold individuals tend to maintain high productivity but at a potential cost to their survival, whereas shy individuals do the opposite [26,69,75]. Studies exploring the fitness consequences of variation in personality largely support this hypothesis, showing that bold individuals often outperform shy ones in terms of reproductive success, but also have reduced survival [26,27,69,76]. In the context of brood parasitism, a trade-off might also exist if bold hosts tend to reject the parasitic egg but at a potential cost stemming from recognition error or punishment by the cuckoo, whereas shy hosts accept (or tolerate) the brood parasitism but suffer reduced reproductive success if the cuckoo egg hatches. Moreover, bold individuals may be more active and hence may be more likely to attract the attention of a cuckoo to their nest, leading to a higher risk of parasitism than in shy (passive) hosts [42]. Higher levels of parasitism would then favour bold hosts to become egg rejecters [18].

Egg-rejection behaviour may be further mediated or directly regulated by physiological mechanisms [77]. For example, a study on American robins *Turdus migratorius* showed that hosts with higher levels of corticosterone, a hormone linked to the stress response, were more likely to reject a parasitic egg [78]. Another study showed that decreasing the levels of prolactin facilitated Eurasian blackbirds *Turdus merula* to reject foreign eggs [79]. The shy-bold continuum may also reflect variation in a range of physiological traits [69,80], including the stress response [81]. For example,

bold (proactive) birds generally have relatively low corticosterone responses whereas shy (reactive) individuals have relatively high corticosterone responses [82,83], but see [84]. However, we did not find a relationship between the probability of egg rejection and breathing rate, a physiological trait related to the stress response [57].

In this study, all personality assays were conducted after the egg-rejection experiments (see Methods). Therefore, hosts that perceived that they were parasitized (and rejected the egg) may have become bolder as a consequence of the treatment. However, this reversal of causation seems less likely, because exploratory behaviour is consistent across years, despite the fact that individuals vary in their experience with artificial or natural brood parasitism.

In conclusion, Daurian redstart females showed strong covariation between exploration and neophobia in the wild, and host personality traits predicted the response to cuckoo parasitism. Specifically, bold hosts (fast-exploring and less neophobic) were more likely to reject a parasitic egg than shy females (slow-exploring and more neophobic). This implies that a cuckoo would have lower success (fitness) when parasitizing a bold host. In the coevolutionary arms race, selection should therefore favour cuckoos that lay their eggs in the nest of a shy host. Whether cuckoos pay attention to the personality of the host or whether bold hosts are more likely to attract a cuckoo warrants further study.

**Ethics.** The experiments comply with the current laws of China, where they were performed. Fieldwork was carried out with permission from Yongji Forestry Bureau, Jilin, China. Experimental procedures were conducted under license from the Animal Management Committee at the College of Life Sciences, Beijing Normal University (permit no. CLS-EAW-2018-001).

**Data accessibility.** Data available from the Dryad Digital Repository: https://doi.org/10.5061/dryad.xwdbrv1cs [85].

**Authors' contributions.** J.Z.: conceptualization, data curation, formal analysis, funding acquisition, investigation, methodology, project administration, resources, software, validation, visualization, writing—original draft, writing—review and editing; P.S.: conceptualization, formal analysis, methodology, software, visualization, writing—review and editing; J.L.: conceptualization, formal analysis, methodology, software, visualization, writing—review and editing; L.Y.: data curation, investigation, resources, writing—review and editing; W.D.: conceptualization, funding acquisition, investigation, methodology, project administration, resources, supervision, validation, writing—review and editing; B.K.: conceptualization, formal analysis, methodology, project administration, supervision, validation, writing—review and editing. All authors gave final approval for publication and agreed to be held accountable for the work performed therein.

**Competing interests.** We declare we have no competing interests.

**Funding.** This study was supported by the National Natural Science Foundation of China grant nos (31372219 and 31672297 to W.D.), and the China Scholarship Council grant no. (201906040159 to J.Z.).

**Acknowledgments.** We are grateful to Eunbi Kwon for her helpful comments with data analysis. We thank Yongji Forestry Bureau, Jilin, China, for their permission and cooperation. We also thank Mengjia Wu, Silvia Quilici, Ziyin Xiong, Rui Wang, Jie Deng, Guobin Zheng and Yangbo Kou for assistance with fieldwork.

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
