## [Peer Review File · Proceedings of the Royal Society B: Biological Sciences]

Review History

RSPB-2021-0228.R0 (Original submission)

Review form: Reviewer 1

Recommendation

Accept with minor revision (please list in comments)

Scientific importance: Is the manuscript an original and important contribution to its field?

Good

General interest: Is the paper of sufficient general interest?

Good

Quality of the paper: Is the overall quality of the paper suitable?

Good

Is the length of the paper justified?

Yes

Should the paper be seen by a specialist statistical reviewer?

No

Do you have any concerns about statistical analyses in this paper? If so, please specify them explicitly in your report.

Yes

It is a condition of publication that authors make their supporting data, code and materials available - either as supplementary material or hosted in an external repository. Please rate, if applicable, the supporting data on the following criteria.

Is it accessible?

No

Is it clear?

N/A

Is it adequate?

N/A

Do you have any ethical concerns with this paper?

No

Comments to the Author

This is a very nice study showing that personality traits (exploration and neophobia) predict some variation in egg rejection in Daurian redstarts. The finding that bold hosts are more likely to reject parasitic/experimental eggs is interesting and suggests that intra-individual variation in hosts requires further study. The paper is very well written, and conclusions are well-supported. It is a valuable contribution to the field.

My main comment is about the statistics. Models with $\Delta AIC < 2$ are compared using Nagelkerke R^2 values (paragraph beginning line 213). The problem with this is that (virtually) always, the addition of a term to a model will improve % variation predicted by the model. Therefore, comparing nested models based on R^2 will result in the least parsimonious model being chosen. Such full models will likely include variables which cannot be generalised beyond your sample (i.e. they will essentially overfit).

I therefore strongly recommend that you compare models differently. Either use a model averaging approach, or select the most parsimonious model from the group of models where $\Delta AIC < 2$, or use likelihood ratio tests to compare nested models. Looking at the supplementary table of all models, all of these options should be reasonable.

Another concern is that you do not distinguish between parasitism causing bold behaviour, or boldness causing increased egg rejection - see comment 4.

Further comments:

1. Abstract: Perhaps specify that the 'parasitic' eggs used in experiments were model eggs.
2. Lines 38-40: Slightly odd choices of references. Several of the references 2-7 would better support the following sentence where 8-10 are referenced (and to a lesser extent vice versa), in my opinion. Please double check that the general references apply well to the points you wish to make.
3. Lines 57-8 and 68-9: Slight contradiction here: is direct evidence absent or scarce? Please clarify. If scarce, cite the relevant study/studies.
4. Methods and Results: As I understand it, all personality assays were conducted after an egg rejection experiment (i.e. when birds had 4-day old chicks or 9-11 days into incubation for exploration and neophobia respectively). It would therefore be worth discussing the direction of causation. Is it not possible that experimental or actual parasitism causes hosts to behave more boldly if they 'know' that they have been parasitised, rather than bold hosts being more likely to reject eggs? How can you tell the difference? This requires some explanation in the discussion section. For example, you may argue that the consistency across years of exploratory behaviour suggests that personality is not labile due to perceived parasitism (but you could not state this for neophobia).
5. Line 124: Do males not reject eggs? Just checking, because if they do, then measuring

personality traits of females is insufficient.

6. Line 239: did this negative relationship still hold for exploratory behaviour and return latency in the second novel object trial? Similarly for other analyses in which return latency in first novel object trial was a variable.
7. Line 249: See my main comment above. This is likely due to comparing models based on R^2 .
8. Line 344: Lovely concluding paragraph!

Review form: Reviewer 2

Recommendation

Major revision is needed (please make suggestions in comments)

Scientific importance: Is the manuscript an original and important contribution to its field?

Good

General interest: Is the paper of sufficient general interest?

Good

Quality of the paper: Is the overall quality of the paper suitable?

Good

Is the length of the paper justified?

Yes

Should the paper be seen by a specialist statistical reviewer?

No

Do you have any concerns about statistical analyses in this paper? If so, please specify them explicitly in your report.

No

It is a condition of publication that authors make their supporting data, code and materials available - either as supplementary material or hosted in an external repository. Please rate, if applicable, the supporting data on the following criteria.

Is it accessible?

Yes

Is it clear?

Yes

Is it adequate?

Yes

Do you have any ethical concerns with this paper?

No

Comments to the Author

This is a very well written paper I enjoyed reading. The stats are thorough and robust. This is an impressive field study with good sample sizes. I do have a couple of concerns.

My four main comments are:

Line 126:

You painted the eggs the same colour, to your eyes, but are the birds seeing the same colour as you are seeing? We know of course that birds see colour different to us. How was the colour matched appropriately within an avian visual sensory capacity?

Line 152:

Ten mins of acclimation seems no time at all for the personality studies? Where birds always instantly retrieved and then have the 10-minute period, or had some birds been waiting longer?

Lines 160-170:

Did this experiment cause nest desertions?

Line 252:

The models overall don't explain much of what was observed. This is not alluded too enough. There needs to be more transparency regarding this. There is a lot going on that isn't explained.

Minor comments

Line 58-60:

The link between these two paragraphs needs better signposting.

Figure 1:

Has not reproduced well and would benefit from some improvements in output quality.

Decision letter (RSPB-2021-0228.R0)

26-Mar-2021

Dear Dr Zhang:

Your manuscript has now been peer reviewed and the reviews have been assessed by an Associate Editor. The reviewers' comments (not including confidential comments to the Editor) and the comments from the Associate Editor are included at the end of this email for your reference. As you will see, the reviewers and the Editors have raised some concerns with your manuscript and we would like to invite you to revise your manuscript to address them.

When submitting your revision please upload a file under "Response to Referees" - in the "File Upload" section. This should document, point by point, how you have responded to the reviewers' and Editors' comments, and the adjustments you have made to the manuscript. We

require a copy of the manuscript with revisions made since the previous version marked as 'tracked changes' to be included in the 'response to referees' document.

Research ethics:

Use of animals and field studies:

It is a condition of publication that you make available the data and research materials supporting the results in the article. Please see our Data Sharing Policies (<https://royalsociety.org/journals/authors/author-guidelines/#data>). Datasets should be deposited in an appropriate publicly available repository and details of the associated accession number, link or DOI to the datasets must be included in the Data Accessibility section of the article (<https://royalsociety.org/journals/ethics-policies/data-sharing-mining/>). Reference(s) to datasets should also be included in the reference list of the article with DOIs (where available).

Online supplementary material will also carry the title and description provided during submission, so please ensure these are accurate and informative. Note that the Royal Society will not edit or typeset supplementary material and it will be hosted as provided. Please ensure that

the supplementary material includes the paper details (authors, title, journal name, article DOI). Your article DOI will be 10.1098/rspb.[paper ID in form xxxx.xxxx e.g. 10.1098/rspb.2016.0049].

Please submit a copy of your revised paper within three weeks. If we do not hear from you within this time your manuscript will be rejected. If you are unable to meet this deadline please let us know as soon as possible, as we may be able to grant a short extension.

Best wishes,
Dr Sasha Dall
mailto: proceedingsb@royalsociety.org

Associate Editor
Board Member: 1

Comments to Author:

This well-written paper presents experimental and correlative data showing that personality traits are associated with the likelihood of cuckoo egg rejection in a host species, the Daurian redstart. This is a nice study system to test hypotheses about inter-individual variation in rejection, since both rejection and acceptance are frequent in the naturally parasitized population. The combination of field experiments using artificial parasitism and tests of exploratory behavior are also impressive. Overall, the results seem solid and both I and the reviewers thought that the paper will make a valuable contribution to the literature. However, there are a number of important concerns that should be addressed in a revision. Reviewer 1 and I both note that using R2 estimates to select a single best-fit model after AIC selection is not statistically sound, and offer alternatives. Although this should not change your main results, it may change the model estimates presented in table 1. Both reviewers also ask for clarifications on important issues, detailed in their comments. My own line-by-line comments are as follows:

17: here and elsewhere, consider rephrasing to avoid convoluted sentences, e.g. "Individuals often vary in their responses..." instead of "there is typically variation in individuals' responses"
Paragraph beginning line 44: the two hypotheses (evolutionary lag vs. equilibrium) are nicely laid out, but the following sentence (beginning "Furthermore") leaves the reader wondering whether these pieces of evidence are meant to support one of those two hypotheses, or whether they suggest additional ones. It seems that some of these factors suggest that the ability to reject eggs is constrained by learning or experience, since older/more experienced birds, and those with perceived high risk of parasitism, are more likely to reject eggs. Is this meant to be interpreted as evidence for a cost/benefit tradeoff where the cost is the risk of making an error?

56-57: here and elsewhere, additional explanation would be useful as to whether, or under what conditions, the host personality hypothesis predicts that variation in egg ejection is adaptive (birds with certain personality traits benefit from rejection whereas those with other traits do not) or not adaptive (perhaps linked to personality traits that are adaptive in other contexts).

85-86: "Being more successful at earlier lines of host defense may decrease selection on later lines of defense." That statement certainly makes sense at the level of the population, but I am not sure that it translates to a tradeoff within individuals, as suggested by the next sentence. It's not clear to me why bold individuals who are better at driving away parasites would necessarily be worse at discriminating parasitic eggs if they are parasitized, unless learning and experience plays a major role in the ability to discriminate?

112-117: are blue egg-laying and pink egg-laying hosts equally likely to be parasitized by cuckoos in this population, or is the risk of parasitism different?

185-191: I am a little concerned about model overfitting for a fairly small sample size in a linear mixed model with one random effect and 5 fixed effects. Did you check for overfitting?

219: Like reviewer 1, I am concerned about using this R2 to select the model with largest explanatory power. These R2 estimates are biased towards models with more parameters, and should not be used in conjunction with an AIC selection approach. You might want to check out Arnold (2010) *J. Wildlife Management*, which essentially argues that of models within 2 AIC

units of the top model, the one with the smallest number of parameters should be the “best fit;” or use model averaging as suggested by Reviewer 1. This should not change your overall conclusions but it might change the estimates of the effect sizes.

245-248: just a comment – how great to have high frequencies of both acceptor and rejectors in the population! Great opportunity to ask this question, perhaps worth emphasizing in the introduction.

Discussion: it is a bit surprising to me that in your models, the personality metrics explain more variation in rejection behavior than egg color does. I would have expected differences between host and egg color to be more salient. Related to this, there is not much explanation of the host egg color polymorphism in the introduction or discussion, which would be helpful since egg color polymorphisms are rare and often interpreted as the result of selection against parasitism. Could you perhaps speculate on this briefly?

Table 1: also please note in the table caption that these represent best-fit models as chosen by AIC, and list all of the parameters that were included in the full models (if they are not all represented here).

Reviewer(s)' Comments to Author:

Referee: 1

Comments to the Author(s)

This is a very nice study showing that personality traits (exploration and neophobia) predict some variation in egg rejection in Daurian redstarts. The finding that bold hosts are more likely to reject parasitic/experimental eggs is interesting and suggests that intra-individual variation in hosts requires further study. The paper is very well written, and conclusions are well-supported. It is a valuable contribution to the field.

My main comment is about the statistics. Models with $\Delta AIC < 2$ are compared using Nagelkerke R^2 values (paragraph beginning line 213). The problem with this is that (virtually) always, the addition of a term to a model will improve % variation predicted by the model. Therefore, comparing nested models based on R^2 will result in the least parsimonious model being chosen. Such full models will likely include variables which cannot be generalised beyond your sample (i.e. they will essentially overfit).

I therefore strongly recommend that you compare models differently. Either use a model averaging approach, or select the most parsimonious model from the group of models where $\Delta AIC < 2$, or use likelihood ratio tests to compare nested models. Looking at the supplementary table of all models, all of these options should be reasonable.

Another concern is that you do not distinguish between parasitism causing bold behaviour, or boldness causing increased egg rejection – see comment 4.

Further comments:

1. Abstract: Perhaps specify that the ‘parasitic’ eggs used in experiments were model eggs.
2. Lines 38-40: Slightly odd choices of references. Several of the references 2-7 would better support the following sentence where 8-10 are referenced (and to a lesser extent vice versa), in my opinion. Please double check that the general references apply well to the points you wish to make.
3. Lines 57-8 and 68-9: Slight contradiction here: is direct evidence absent or scarce? Please clarify. If scarce, cite the relevant study/studies.
4. Methods and Results: As in understand it, all personality assays were conducted after an egg rejection experiment (i.e. when birds had 4-day old chicks or 9-11 days into incubation for exploration and neophobia respectively). It would therefore be worth discussing the direction of causation. Is it not possible that experimental or actual parasitism causes hosts to behave more boldly if they ‘know’ that they have been parasitised, rather than bold hosts being more likely to reject eggs? How can you tell the difference? This requires some explanation in the discussion section. For example, you may argue that the consistency across years of exploratory behaviour suggests that personality is not labile due to perceived parasitism (but you could not state this for neophobia).
5. Line 124: Do males not reject eggs? Just checking, because if they do, then measuring personality traits of females is insufficient.

6. Line 239: did this negative relationship still hold for exploratory behaviour and return latency in the second novel object trial? Similarly for other analyses in which return latency in first novel object trial was a variable.
7. Line 249: See my main comment above. This is likely due to comparing models based on R^2 .
8. Line 344: Lovely concluding paragraph!

Referee: 2

Comments to the Author(s)

This is a very well written paper I enjoyed reading. The stats are thorough and robust. This is an impressive field study with good sample sizes. I do have a couple of concerns.

My four main comments are:

Line 126:

You painted the eggs the same colour, to your eyes, but are the birds seeing the same colour as you are seeing? We know of course that birds see colour different to us. How was the colour matched appropriately within an avian visual sensory capacity?

Line 152:

Ten mins of acclimation seems no time at all for the personality studies? Where birds always instantly retrieved and then have the 10-minute period, or had some birds been waiting longer?

Lines 160-170:

Did this experiment cause nest desertions?

Line 252:

The models overall don't explain much of what was observed. This is not alluded too enough. There needs to be more transparency regarding this. There is a lot going on that isn't explained.

Minor comments

Line 58-60:

The link between these two paragraphs needs better signposting.

Figure 1:

Has not reproduced well and would benefit from some improvements in output quality.

Author's Response to Decision Letter for (RSPB-2021-0228.R0)

See Appendices A & B.

Decision letter (RSPB-2021-0228.R1)

17-May-2021

Dear Professor Deng

I am pleased to inform you that your manuscript RSPB-2021-0228.R1 entitled "Host personality predicts cuckoo egg rejection in Daurian redstarts *Phoenicurus aureus*" has been accepted for publication in Proceedings B.

The AE has recommended publication, but also suggests some minor revisions to your manuscript. Therefore, I invite you to respond to the comments and revise your manuscript. Because the schedule for publication is very tight, it is a condition of publication that you submit the revised version of your manuscript within 7 days. If you do not think you will be able to meet this date please let us know.

In order to ensure effective and robust dissemination and appropriate credit to authors the dataset(s) used should be fully cited. To ensure archived data are available to readers, authors should include a 'data accessibility' section immediately after the acknowledgements section.

This should list the database and accession number for all data from the article that has been made publicly available, for instance:

[http://datadryad.org/submit?journalID=RSPB&manu=\(Document not available\)](http://datadryad.org/submit?journalID=RSPB&manu=(Document%20not%20available)) which will take you to your unique entry in the Dryad repository. If you have already submitted your data to dryad you can make any necessary revisions to your dataset by following the above link. Please see <https://royalsociety.org/journals/ethics-policies/data-sharing-mining/> for more details.

Sincerely,
Dr Sasha Dall
Editor, Proceedings B
<mailto:proceedingsb@royalsociety.org>

Associate Editor:

Comments to Author:

Thanks to the authors for nicely revising their manuscript to take my and the reviewers' suggestions into account. The resulting manuscript reads nicely and the statistical analyses are more straightforward. I have only one very small remaining comment, which could perhaps be incorporated at the proofs stage.

I had previously asked whether blue egg-laying and pink egg-laying hosts are equally likely to be parasitized by cuckoos in this population. You provide a good response in the response to referees, but I think it would be useful to include a very brief explanation (one sentence or less) in the main text as well. This was your reply:

"In our study population, redstarts laying blue eggs suffered higher risk of parasitism than hosts laying pink eggs. However, hosts laying pink eggs are also more likely to reject parasitic eggs, and it is therefore not necessarily easy to obtain unbiased information on parasitizing rates (as hosts may have ejected the parasitic egg before we detected it). We therefore do not yet know for sure whether cuckoos preferentially target blue clutches. This is currently subject of further investigation."

Otherwise, I have no further comments on this nice study.

Author's Response to Decision Letter for (RSPB-2021-0228.R1)

See Appendices C & D.

Decision letter (RSPB-2021-0228.R2)

19-May-2021

Dear Professor Deng

I am pleased to inform you that your manuscript entitled "Host personality predicts cuckoo egg rejection in Daurian redstarts *Phoenicurus aureus*" has been accepted for publication in Proceedings B.

Your article has been estimated as being 9 pages long. Our Production Office will be able to confirm the exact length at proof stage.

Data Accessibility section

Open Access

Paper charges

You are allowed to post any version of your manuscript on a personal website, repository or preprint server. However, the work remains under media embargo and you should not discuss it

with the press until the date of publication. Please visit <https://royalsociety.org/journals/ethics-policies/media-embargo> for more information.

Sincerely,
Proceedings B
<mailto:proceedingsb@royalsociety.org>

Appendix A

June 04, 2021

To Prof. Dr. Sasha Dall

Editor Proceedings of the Royal Society London B

Re revision MS RSPB-2021-0228

Dear Prof. Dall,

Thank you very much for giving us the opportunity to revise our manuscript. We have read the comments carefully and have revised the manuscript as suggested by the associate editor and reviewers. Please find herewith the revised manuscript and our detailed responses to all the comments. Line numbers in the response document refer to the highlighted version of the manuscript. We hope the associate editor and reviewers will find that the manuscript has been significantly improved.

Thank you very much for your assistance. Please do not hesitate to contact us if there are any other questions or issues we should address.

Yours sincerely, on behalf of the co-authors,

Jinggang Zhang

Appendix B

Response to Comments Associate Editor and Referees (RSPB-2021-0228)

Associate Editor

Board Member: 1

Comments to Author:

This well-written paper presents experimental and correlative data showing that personality traits are associated with the likelihood of cuckoo egg rejection in a host species, the Daurian redstart. This is a nice study system to test hypotheses about inter-individual variation in rejection, since both rejection and acceptance are frequent in the naturally parasitized population. The combination of field experiments using artificial parasitism and tests of exploratory behavior are also impressive. Overall, the results seem solid and both I and the reviewers thought that the paper will make a valuable contribution to the literature. However, there are a number of important concerns that should be addressed in a revision. Reviewer 1 and I both note that using R2 estimates to select a single best-fit model after AIC selection is not statistically sound, and offer alternatives. Although this should not change your main results, it may change the model estimates presented in table 1. Both reviewers also ask for clarifications on important issues, detailed in their comments.

Reply:

Thank you very much for the positive appraisal of our study. Re the main points: see our responses below.

My own line-by-line comments are as follows:

17: here and elsewhere, consider rephrasing to avoid convoluted sentences, e.g. “Individuals often vary in their responses...” instead of “there is typically variation in individuals’ responses”

Reply:

Thank you for your helpful suggestion. We have rephrased the relevant sentences (lines 17-18, 115-116).

Paragraph beginning line 44: the two hypotheses (evolutionary lag vs. equilibrium) are nicely laid out, but the following sentence (beginning “Furthermore”) leaves the reader wondering whether these pieces of evidence are meant to support one of those two hypotheses, or whether they suggest additional ones. It seems that some of these factors suggest that the ability to reject eggs is constrained by learning or experience, since older/more experienced birds, and those with perceived high risk of parasitism, are more likely to reject eggs. Is this meant to be interpreted as evidence for a cost/benefit tradeoff where the cost is the risk of making an error?

Reply:

We greatly appreciate this comment. In this study, we focused on variation in egg rejection

at the individual level, while the two hypotheses (evolutionary lag vs. equilibrium) are about the population level. Because this is not so relevant to our focus, we decided to remove the section about the two hypotheses and we revised the paragraph to clarify the issue (lines 45-71).

56-57: here and elsewhere, additional explanation would be useful as to whether, or under what conditions, the host personality hypothesis predicts that variation in egg ejection is adaptive (birds with certain personality traits benefit from rejection whereas those with other traits do not) or not adaptive (perhaps linked to personality traits that are adaptive in other contexts).

Reply:

We now clarify that variation in egg-rejection behaviour between individuals with different personality types can be adaptive if birds with different personalities also differ in the risk of being parasitized (lines 73-74).

85-86: "Being more successful at earlier lines of host defense may decrease selection on later lines of defense." That statement certainly makes sense at the level of the population, but I am not sure that it translates to a tradeoff within individuals, as suggested by the next sentence. It's not clear to me why bold individuals who are better at driving away parasites would necessarily be worse at discriminating parasitic eggs if they are parasitized, unless learning and experience plays a major role in the ability to discriminate?

Reply:

The idea is that bold individuals may be less efficient at discriminating and rejecting the parasitic eggs, because of the higher success in driving away adult parasites, and therefore the reduced opportunity to learn to reject eggs. We have now clarified this (line 109).

112-117: are blue egg-laying and pink egg-laying hosts equally likely to be parasitized by cuckoos in this population, or is the risk of parasitism different?

Reply:

In our study population, redstarts laying blue eggs suffered higher risk of parasitism than hosts laying pink eggs. However, hosts laying pink eggs are also more likely to reject parasitic eggs, and it is therefore not necessarily easy to obtain unbiased information on parasitizing rates (as hosts may have ejected the parasitic egg before we detected it). We therefore do not yet know for sure whether cuckoos preferentially target blue clutches. This is currently subject of further investigation.

185-191: I am a little concerned about model overfitting for a fairly small sample size in a linear mixed model with one random effect and 5 fixed effects. Did you check for overfitting?

Reply:

Thank you for pointing out this potential issue. To assess whether overfitting is a problem,

we first ran LMMs to determine which variables explained significant variation in each personality trait. We then reran the repeatability models only including the variables that explained significant variation in each personality trait. For example, only the interval between two tests and the date of the test explained significant variation in exploratory behaviour (see supplementary table S1), and we therefore only included these two variables when calculating repeatability of exploratory behaviour. Because this did not qualitatively affect the repeatability estimates, we kept the original estimates in the main text, but referred to the estimates with the reduced models in the supplement. We have revised all relevant sections of the manuscript (lines 228-234). Note that we are not interested in the significance of the fixed effects, but rather want to control for variables that might obscure the “true” repeatability.

219: Like reviewer 1, I am concerned about using this R2 to select the model with largest explanatory power. These R2 estimates are biased towards models with more parameters, and should not be used in conjunction with an AIC selection approach. You might want to check out Arnold (2010) J. Wildlife Management, which essentially argues that of models within 2 AIC units of the top model, the one with the smallest number of parameters should be the “best fit;” or use model averaging as suggested by Reviewer 1. This should not change your overall conclusions but it might change the estimates of the effect sizes.

Reply:

We greatly appreciate this comment. We now follow the suggestion to always select the most parsimonious model (i.e. the model with the smallest number of parameters) among those with a ΔAIC_c lower than 2 (see below and lines 255-259).

245-248: just a comment – how great to have high frequencies of both acceptor and rejectors in the population! Great opportunity to ask this question, perhaps worth emphasizing in the introduction.

Reply:

Thank you very much. We have added a sentence in the introduction to emphasize this point (lines 116-118).

Discussion: it is a bit surprising to me that in your models, the personality metrics explain more variation in rejection behavior than egg color does. I would have expected differences between host and egg color to be more salient. Related to this, there is not much explanation of the host egg color polymorphism in the introduction or discussion, which would be helpful since egg color polymorphisms are rare and often interpreted as the result of selection against parasitism. Could you perhaps speculate on this briefly?

Reply:

We are grateful for this excellent suggestion. Because we now adopt a different model selection approach, clutch colour is not in the final model examining return latency anymore. In the other two models, clutch colour is retained, and in these models it actually

explains more variation than any of the other variables. We have added a section in the introduction about the role of the egg colour polymorphism in egg-rejection behaviour in Daurian redstarts (lines 119-122).

Table 1: also please note in the table caption that these represent best-fit models as chosen by AIC, and list all of the parameters that were included in the full models (if they are not all represented here).

Reply:

Changed as suggested (lines 674-677).

Reviewer(s)' Comments to Author:

Referee: 1

Comments to the Author(s)

This is a very nice study showing that personality traits (exploration and neophobia) predict some variation in egg rejection in Daurian redstarts. The finding that bold hosts are more likely to reject parasitic/experimental eggs is interesting and suggests that intra-individual variation in hosts requires further study. The paper is very well written, and conclusions are well-supported. It is a valuable contribution to the field.

My main comment is about the statistics. Models with $\Delta AIC < 2$ are compared using Nagelkerke R^2 values (paragraph beginning line 213). The problem with this is that (virtually) always, the addition of a term to a model will improve % variation predicted by the model. Therefore, comparing nested models based on R^2 will result in the least parsimonious model being chosen. Such full models will likely include variables which cannot be generalised beyond your sample (i.e. they will essentially overfit).

I therefore strongly recommend that you compare models differently. Either use a model averaging approach, or select the most parsimonious model from the group of models where $\Delta AIC < 2$, or use likelihood ratio tests to compare nested models. Looking at the supplementary table of all models, all of these options should be reasonable.

Reply:

Thank you very much for the positive appraisal of our study and for the helpful comments. We agree with your point regarding the model selection, and followed your suggestion to always select the most parsimonious model (i.e. the model with the smallest number of parameters) with a $\Delta AICc$ lower than 2 (lines 255-259). We have revised all relevant sections of the manuscript accordingly (lines 290-300, table 1).

Another concern is that you do not distinguish between parasitism causing bold behaviour, or boldness causing increased egg rejection – see comment 4.

Reply:

Thank you for this interesting point. We provide a detailed response on this issue below.

Further comments:

1. Abstract: Perhaps specify that the ‘parasitic’ eggs used in experiments were model eggs.

Reply:

In this study, we used both model eggs and real cuckoo eggs to conduct the egg-rejection experiment. We have now clarified this (lines 159-160, 165-166). We therefore prefer to keep the term ‘parasitic egg’ in the Abstract, as it is more appropriate than ‘model egg’.

2. Lines 38-40: Slightly odd choices of references. Several of the references 2-7 would better support the following sentence where 8-10 are referenced (and to a lesser extent vice versa), in my opinion. Please double check that the general references apply well to the points you wish to make.

Reply:

We agree and have changed the references (lines 41-42).

3. Lines 57-8 and 68-9: Slight contradiction here: is direct evidence absent or scarce? Please clarify. If scarce, cite the relevant study/studies.

Reply:

Thanks for pointing this out. To our knowledge, there is only one study testing the relationship between host personality and egg rejection, but it did not find any relationship. There is thus some previous work on the relation between personality and egg-rejection behaviour, but no evidence supporting the host personality hypothesis. We have now clarified this and added a citation to the relevant study (lines 87-89).

4. Methods and Results: As in understand it, all personality assays were conducted after an egg rejection experiment (i.e. when birds had 4-day old chicks or 9-11 days into incubation for exploration and neophobia respectively). It would therefore be worth discussing the direction of causation. Is it not possible that experimental or actual parasitism causes hosts to behave more boldly if they ‘know’ that they have been parasitised, rather than bold hosts being more likely to reject eggs? How can you tell the difference? This requires some explanation in the discussion section. For example, you may argue that the consistency across years of exploratory behaviour suggests that personality is not labile due to perceived parasitism (but you could not state this for neophobia).

Reply:

Thank you very much for this interesting comment and suggestion. As the reviewer pointed out, the high repeatability of exploratory behaviour suggests that it is a consistent trait across years. Please, also note that some individuals did not experience artificial or natural brood parasitism, so it seems somewhat unlikely that the parasitism affected the exploratory behaviour. Nevertheless, we have added a section in the discussion where we mention this possibility (lines 390-394).

5. Line 124: Do males not reject eggs? Just checking, because if they do, then measuring personality traits of females is insufficient.

Reply:

Good point. Males do not reject eggs in Daurian redstarts, only females do. We have clarified this (lines 116-117).

6. Line 239: did this negative relationship still hold for exploratory behaviour and return latency in the second novel object trial? Similarly for other analyses in which return latency in first novel object trial was a variable.

Reply:

The negative relationship still held for exploratory behaviour and return latency in the second novel object trial ($r_s = -0.40$, $p = 0.04$, $n = 27$). The relationship between egg-rejection behaviour and return latency in the second novel object trial was also significant (GLM: estimate = -2.36, SE = 1.15, $z = -2.05$, $p = 0.04$).

7. Line 249: See my main comment above. This is likely due to comparing models based on R^2 .

Reply:

We are grateful for this comment. We followed the reviewer's suggestion and selected the most parsimonious model with a $\Delta AICc$ lower than 2. We have revised all relevant sections accordingly (lines 255-259, 290-300, table 1). Please note that this did not change any of the conclusions.

8. Line 344: Lovely concluding paragraph!

Reply:

We appreciate the positive feedback.

Referee: 2

Comments to the Author(s)

This is a very well written paper I enjoyed reading. The stats are thorough and robust. This is an impressive field study with good sample sizes. I do have a couple of concerns.

Reply:

Thank you very much for the positive appraisal of our study and for the helpful comments.

My four main comments are:

Line 126:

You painted the eggs the same colour, to your eyes, but are the birds seeing the same colour as you are seeing? We know of course that birds see colour different to us. How was the colour matched appropriately within an avian visual sensory capacity?

Reply:

Like most studies that performed egg-rejection experiments (Yang et al, 2015, 2016; Moskát et al, 2014; Samaš et al, 2016; Thomson et al, 2016), we made model eggs from clay. To the human eye, the colour of model eggs is very similar to that of real cuckoo eggs, but we did not measure whether the colour matched appropriately given the avian vision system. Thus, indeed we do not know whether the model eggs also look the same as real cuckoo eggs to birds. However, this should not affect the validity of the experiment, because redstarts behaved in a similar way to the model eggs as to real cuckoo eggs.

Line 152:

Ten mins of acclimation seems no time at all for the personality studies? Where birds always instantly retrieved and then have the 10-minute period, or had some birds been waiting longer?

Reply:

Our pilot experiments showed that nearly all birds settled down within 10 min, so we set a 10-min period of acclimatization a priori (before the experiments). This period is longer than that used in some similar studies on exploratory behaviour. For example, Stuber et al. (2013) only acclimated great tits for 1 min before the experiment.

Birds were always retrieved immediately after being caught, normally within 1 min. After extracting birds from the mist-net, tuck net, spring net traps or glue stick, we measured and banded them immediately, which usually took about 5 min. After that, we put the birds into the small chamber to acclimate.

Lines 160-170:

Did this experiment cause nest desertions?

Reply:

We previously showed that nest desertion rates do not differ between experimentally

parasitized and non-manipulated control nests (Zhang et al. 2021). Therefore, although there was a small number of deserted nests in this study ($n = 9$), we do not believe this was related to the experimental procedure (see also lines 174-177).

Line 252:

The models overall don't explain much of what was observed. This is not alluded to enough. There needs to be more transparency regarding this. There is a lot going on that isn't explained.

Reply:

The explanatory power of our models (R^2 : 0.30-0.42) is substantial, and larger than or similar to other studies (Hanley et al. 2017,2019; Stoddard et al. 2019). However, we agree with the reviewer that there is considerable remaining variation and that further work on egg rejection behaviour in Daurian redstarts would therefore be valuable. We have now added a section in the discussion where we highlight this (lines 327-329).

Minor comments

Line 58-60:

The link between these two paragraphs needs better signposting.

Reply:

We have modified this such that the transition between these two paragraphs is smoother (lines 69-76).

Figure 1:

Has not reproduced well and would benefit from some improvements in output quality.

Reply:

We have increased the resolution of the predicted values and 95% CI, which we hope improved the appearance of the figure.

Appendix C

June 04, 2021

To Prof. Dr. Sasha Dall

Editor Proceedings of the Royal Society London B

Re revision MS RSPB-2021-0228

Dear Prof. Dall,

Thank you very much for accepting our manuscript (RSPB-2021-0228). We have read the comments carefully and have revised the manuscript accordingly. Please find herewith the revised manuscript and our response to the comment. Line numbers in the response document refer to the highlighted version of the manuscript.

Thank you very much for your assistance. Please do not hesitate to contact us if there are any other questions or issues we should address.

Yours sincerely, on behalf of the co-authors,

Jinggang Zhang

Appendix D

Response to Referees (RSPB-2021-0228)

Associate Editor:

Comments to Author:

Thanks to the authors for nicely revising their manuscript to take my and the reviewers' suggestions into account. The resulting manuscript reads nicely and the statistical analyses are more straightforward. I have only one very small remaining comment, which could perhaps be incorporated at the proofs stage.

I had previously asked whether blue egg-laying and pink egg-laying hosts are equally likely to be parasitized by cuckoos in this population. You provide a good response in the response to referees, but I think it would be useful to include a very brief explanation (one sentence or less) in the main text as well. This was your reply:

"In our study population, redstarts laying blue eggs suffered higher risk of parasitism than hosts laying pink eggs. However, hosts laying pink eggs are also more likely to reject parasitic eggs, and it is therefore not necessarily easy to obtain unbiased information on parasitizing rates (as hosts may have ejected the parasitic egg before we detected it). We therefore do not yet know for sure whether cuckoos preferentially target blue clutches. This is currently subject of further investigation."

Otherwise, I have no further comments on this nice study.

Reply:

Thank you very much for the positive appraisal of our work and the constructive suggestion. We have added a sentence to emphasize this point (Lines 105-108).